# Post-COVID-19 Syndrome in Non-Hospitalized Individuals: Healthcare Situation 2 Years after SARS-CoV-2 Infection

**DOI:** 10.3390/v15061326

**Published:** 2023-06-05

**Authors:** Inge Kirchberger, Christine Meisinger, Tobias D. Warm, Alexander Hyhlik-Dürr, Jakob Linseisen, Yvonne Goßlau

**Affiliations:** 1Epidemiology, Faculty of Medicine, University of Augsburg, 86156 Augsburg, Germany; christine.meisinger@med.uni-augsburg.de (C.M.); jakob.linseisen@med.uni-augsburg.de (J.L.); 2Vascular Surgery, Faculty of Medicine, University of Augsburg, 86156 Augsburg, Germany; tobias.warm@uk-augsburg.de (T.D.W.); alexander.hyhlik-duerr@uk-augsburg.de (A.H.-D.); yvonne.gosslau@uk-augsburg.de (Y.G.); 3Institute for Medical Information Processing, Biometry and Epidemiology–IBE, LMU Munich, 81377 Munich, Germany

**Keywords:** outpatients, healthcare, COVID-19, long COVID, post-COVID

## Abstract

Although “post-COVID-19 syndrome” (PCS) is reported to be common even in non-hospitalized individuals, long-term information on symptom burden, healthcare needs, utilization, and satisfaction with healthcare is scarce. The objectives of this study were to describe symptom burden, healthcare utilization and experiences with the healthcare offered for PCS in a German sample of non-hospitalized persons 2 years after SARS-CoV-2 infection. Individuals with past COVID-19 confirmed by positive polymerase chain reaction testing were examined at the University Hospital of Augsburg from 4 November 2020 to 26 May 2021 and completed a postal questionnaire between 14 June 2022 and 1 November 2022. Participants who self-reported the presence of fatigue, dyspnea on exertion, memory problems or concentration problems were classified as having PCS. Of the 304 non-hospitalized participants (58.2% female, median age 53.5), 210 (69.1%) had a PCS. Among these, 18.8% had slight to moderate functional limitations. Participants with PCS showed a significantly higher utilization of healthcare and a large proportion complained about lacking information on persistent COVID-19 symptoms and problems finding competent healthcare providers. The results indicate the need to optimize patient information on PCS, facilitate access to specialized healthcare providers, provide treatment options in the primary care setting and improve the education of healthcare providers.

## 1. Introduction

A considerable proportion of patients infected with the coronavirus SARS-CoV-2 experience symptoms such as fatigue, dyspnea, and cognitive problems that persist several weeks or months after the acute coronavirus disease 2019 (COVID-19) [1,2,3,4]. This long-term sequelae is commonly called “long COVID” or “post-COVID syndrome/condition” [5,6]. The proportion of persons affected with post-COVID syndrome (PCS), which includes persistence of symptoms for at least 12 weeks, varies depending on the specific definition of PCS, the study design and symptom assessment, and the severity of the acute COVID-19, and ranges between 6% and 46% in non-hospitalized persons [3,4,7,8,9]. From the healthcare system point of view, its prevalence in non-hospitalized persons is particularly important because this group makes up 80% [10] to 97% [8] of all COVID-19 cases. The large number of persons with PCS challenges the healthcare systems since many of them may require specific treatment and support [11].

So far, scientific investigations on the healthcare needs, utilization and the patients’ experiences have been scarce. Qualitative studies in different countries found difficulties in accessing healthcare services for PCS and showed that experiences with healthcare providers, services and systems as well as the challenges of obtaining appropriate information were a major concern of the affected persons [12,13,14,15,16,17]. An online survey of 2113 persons with long COVID in the Netherlands and Belgium identified various unmet specific information needs and a large number of persons who were dissatisfied with COVID-19 aftercare [18]. The only study from Germany consisted of an online survey of 126 persons with long COVID and a postal survey of 73 general practitioners [19]. Heterogenous ratings of satisfaction with medical care and attitudes towards patients and their disease were found, and patient and healthcare practitioners suggested a structured concept of care with competent contact points and good coordination of healthcare [19].

Furthermore, most studies having investigated PCS have a follow-up (FUP) time of less than 2 years. However, long-term information is needed to assure that healthcare services appropriately consider the specific short- and long-term needs of individuals with PCS.

Thus, the objectives of the present study were to characterize the symptom burden in a German sample of non-hospitalized persons 2 years after SARS-CoV-2 infection and to describe their healthcare utilization and experiences with the healthcare offered for persistent COVID-19 symptoms.

## 2. Materials and Methods

### 2.1. Design and Study Population

The present study is a follow-up assessment of the Corona Thrombosis Study (COVID-T), a prospective single-center observational study evaluating the consequences of COVID-19 for the vascular system [20,21,22]. The study sample was recruited from the population living in the city and the county of Augsburg. The public health departments identified eligible persons with past COVID-19 confirmed by positive polymerase chain reaction (PCR) testing and sent out a total of 1600 postal invitations for study participation between 21 October 2020 and 6 November 2020. The potential study participants were invited to clinical examinations and assessments that were performed at the University Hospital of Augsburg from 4 November 2020 to 26 May 2021. A total of 525 (32.8%) participants were enrolled in the study. A postal follow-up survey was conducted between 14 June 2022 and 1 November 2022. Of the 525 persons, 361 (69%) returned a completed questionnaire. The present analysis is based on 304 persons who were not hospitalized for their initial COVID-19 disease (see Appendix A).

The study was approved by the ethics committee of the Ludwig-Maximilians Universität Munich and was performed in accordance with the Declaration of Helsinki. Written informed consent was obtained from all participants.

### 2.2. Measures

Data were collected using a self-reporting questionnaire which was administered on a tablet personal computer at the baseline examination and on paper in the postal follow-up survey. The questionnaire covered information on socio-demographics, disease history, comorbid conditions as well as symptoms during the acute COVID-19 infection and persisting symptoms. The participants were asked to complete a self-developed list of 42 symptoms, rating them for their occurrence in the acute COVID-19 phase as well as for the 14 days before the baseline examination and the follow-up survey.

In the follow-up postal survey, the participants were additionally asked about health care utilization in the past 4 weeks and 12 months, and responded to eight questions on experiences regarding PCS health care and nine questions on experiences endured with long-lasting fatigue. Moreover, functional limitations were assessed using the Post-COVID Functional Status Questionnaire (PCFS) [23].

### 2.3. Definition of PCS

In the present study, a definition of the PCS largely based on the World Health Organization (WHO) clinical case definition [5] was applied: Participants who self-reported the presence of fatigue, dyspnea on exertion, memory problems or concentration problems, either at the baseline assessment (median 9 months after acute infection) or in the follow-up (median 26 months after acute infection), were classified as having PCS.

### 2.4. Data Analysis

A Chi square test or Fisher’s exact test was used to determine differences between persons with or without PCS in nominal variables and a Mann–Whitney U-Test in ordinal variables, respectively. For statistical tests, an alpha level of 0.05 was defined. Statistical analyses were performed using SAS Version 9.4 (SAS Institute, Cary, NC, USA).

## 3. Results

### 3.1. Sample Characteristics

The study sample consisted of 177 (58.2%) women and 127 (41.8%) men with a median age of 53 years (IQR 41; 61). Further characteristics are detailed in Table 1.

Of the 304 participants, 183 (60.2%) agreed to having experienced COVID-19 related symptoms longer than 4 weeks, and 138 out of 303 (45.5%) longer than 3 months, respectively. Based on the report of COVID-19 symptoms, 210 (69.1%) were classified as having PCS. Among these, 63 (30.1%) perceived themselves as suffering from PCS, 60 (28.7%) were unsure, and 86 (41.2%) stated not having PCS.

### 3.2. COVID-19 Symptoms

At least one symptom was reported by 245 participants (80.6%) at the baseline and 262 persons (86.2%) at the FUP. Fatigue was the most common symptom at the baseline (33.9%) and the FUP (52.8%), followed by muscle or joint pain (22.0%, 42.1%), headache (25.1%, 37.6%), concentration problems (27.2%, 34.9%) and memory problems (23.4%, 33.6%). Dyspnea on exertion was reported by 24.4% of the participants at the baseline and 27.4% at the FUP (see Appendix A). With the exception of impairment of smell or taste functions and heartburn, all symptoms were more common at the FUP than at the baseline.

Participants with PCS had significantly higher prevalences in 33 out of 42 symptoms assessed at the FUP (see Table 2). In addition, the median number of symptoms at the baseline was 6 (IQR 3; 10) in persons with PCS and 1 (0; 2) in persons without PCS. At the FUP, persons with PCS had a median of 9 (5; 15) symptoms compared with persons without PCS who had a median of 1 (0; 3) symptom. Differences at both time points were significant (*p* < 0.0001).

### 3.3. Healthcare Utilization

Table 3 shows that general practitioners were most often attended in the past 4 weeks, followed by several medical specialists and physical therapists. Specialists in psychiatry/psychotherapy were significantly more often attended by persons with PCS. In the past year, most of the medical specialists as well as physical therapists, psychologists/psychotherapists and non-medical practitioners were significantly more often visited by persons with PCS.

### 3.4. Experiences with PCS Healthcare

Among all participants, 143 (48.5%) were dissatisfied with the information on PCS provided by the media, 118 (40.7%) with the information through physicians/therapists, and 162 (61.1%) with the information through health insurance companies and other healthcare providers.

Among those who confirmed having experienced COVID-19-related symptoms for at least 3 months after diagnosis (*n* = 138), 85/129 (64.8%) reported difficulties in finding an appropriate point of contact for their complaints, 89/129 (68.9%) in finding good information about long-lasting COVID-19 symptoms, and 57/130 (43.8%) reported that information about long-lasting complaints following COVID-19 was mostly unclear and difficult to understand (see Figure 1). Support and understanding from others, including health professionals, was considered less problematic.

Most of the 123 participants (40.5%) who reported having experienced fatigue for more than 3 months following the acute COVID-19 event confirmed that fatigue was specifically severe after vigorous exercise or mental strain (*n* = 96, 78.7%) and 64 (56.2%) perceived fatigue as the worst consequence of the COVID-19 disease (see Figure 2). In addition, almost one half of the persons reported helplessness regarding their fatigue and problems in receiving professional support.

## 4. Discussion

The present study found that even 2 years after SARS-CoV-2 infection, affected persons with a mild disease course had a number of persisting symptoms and 69.1% can be classified as having PCS. Among these, 18.8% had slight to moderate functional limitations. Study participants with PCS had a significantly higher utilization of healthcare and a large proportion complained about lacking information on long-lasting COVID-19 symptoms and problems in finding competent healthcare providers.

In general, the frequency of COVID-19 related symptoms was higher at 26 months after the onset of the disease than at 9 months after, with the exception of impairment to the sense of taste or smell, and heartburn. This is in line with another German study showing an increase in the prevalence of fatigue and dyspnea from 5 to 12 months post-COVID [24] and with studies reporting that olfactory dysfunction disappears in most patients over time [25]. However, the fact that most symptoms persisted over 2 years indicates the need to further investigate the long-term course of PCS and factors contributing to an improvement or deterioration in symptoms. Furthermore, the persistence of PCS in persons with mild COVID-19 courses in the present study suggests that a considerable proportion of the population may need medical care for their PCS-related health problems over a long period of time. Healthcare providers should be prepared to manage these challenges and the healthcare system should offer additional resources to support healthcare providers and affected persons.

Indeed, the present study showed that a number of healthcare providers were involved in the healthcare of individuals with PCS in the second year after the onset of the disease more often than in persons without PCS. An overall increase in the utilization of healthcare services was also found in a German study comparing persons with confirmed post-acute COVID-19 (using the diagnostic code) and a control group without COVID-19 diagnosis based on nationwide claims data [26]; similar findings were reported in a study from Israel [27]. General practitioners were the major point of contact for persons with PCS in the present study. Schulz et al. [26] also reported that three out of four patients diagnosed with post-acute COVID-19 exclusively received treatment from a primary care physician, specifically referring to a problem-oriented discussion. This highlights the important role of primary care providers as a first point of contact and in the coordination of patient care over time.

Only 4.8% of the individuals with PCS made use of specialized COVID clinics. Reasons for the non-utilization of these clinics may include a low symptom burden and absence of functional impairments, lacking information on these healthcare facilities or difficulties in obtaining an appointment. The study participants’ responses to the questions on satisfaction with information and treatment indicate a lack of appropriate information on long-lasting COVID-19 symptoms and a lack of support from the healthcare system for more than one half of the participants. Largely comparable results were found in a previous German study [19].

Overall, the large variety of symptoms and involved medical disciplines suggest multidisciplinary models of healthcare coordinated by general practitioners and applying a stepped-care approach [19], mobile primary healthcare for patients in rural areas [28] and digital interventions for individuals with minor complaints [29].

Interestingly, we found a mismatch between the applied definition of PCS and the persons’ subjective perception of having PCS. Most of those who were classified as having PCS did not share this view or were unsure. Possibly, persons with a larger number of symptoms, a higher symptom severity or more functional limitations are more likely to perceive themselves as having PCS than persons with a few mild symptoms. In addition, psychological and social factors may influence a person’s perception of having an illness. Against this background, on the one hand, the broad PCS definitions restricted to persisting symptoms and resulting in a large number of affected persons may be useful for offering healthcare to everyone who needs support. On the other hand, many persons who do not feel strongly impaired by their symptoms would be labeled as being ill. Overall, it seems crucial that diagnosis and treatment of PCS are based on the bio-psycho-social disease model and also consider the impact of the individual’s psychosocial background [30,31]. Futhermore, a possible benefit of extended definitions of PCS that include functional impairment and health-related quality of life should be discussed, in order to avoid classifying people with minor symptoms as ill and in need of treatment [32]. The current WHO definition based on an expert Delphi procedure already mentions that symptoms “generally have an impact on everyday life” [5].

To our knowledge, this is the first study which is based on a two-year follow-up of non-hospitalized persons with COVID-19 investigating PCS and healthcare utilization in Germany. Only persons with confirmed positive PCR testing were included in the study. A limitation which applies to all studies investigating PCS is the lack of a common definition of long COVID and PCS. This limits the comparability of results across studies. Furthermore, the proportion of persons with PCS may be overestimated because persons who participated in both surveys may have experienced a higher symptom burden than those who rejected participation. In addition, psychosocial factors and the growing media attention on PCS may have influenced the report of symptoms [33]. Healthcare utilization was assessed retrospectively and the questions’ timeframe did not cover the first months after the onset of the disease.

## 5. Conclusions

Overall, the results of the present study highlight the need to (1) optimize patient information on PCS and the most common symptoms such as fatigue, (2) faciliate access to specialized healthcare providers and to easily accessible treatment options coordinated by primary care specialists, and (3) improve the education of healthcare providers on PCS. Further long-term studies are required to gain comprehensive knowledge on the course of PCS, the perceptions and needs of the affected individuals, and how the healthcare system can meet these needs.

## Figures and Tables

**Figure 1 viruses-15-01326-f001:**
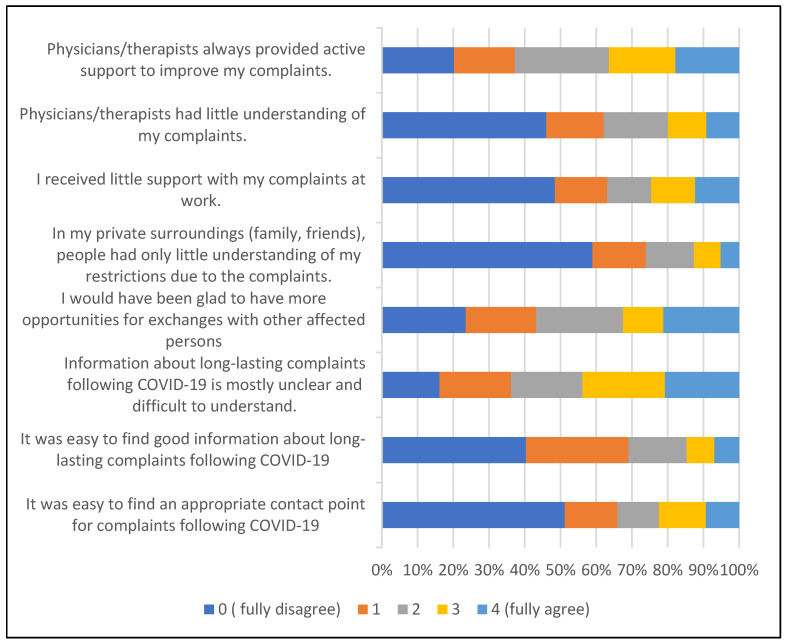
Experiences of having COVID-19-related symptoms for at least 3 months after diagnosis (*n* = 138).

**Figure 2 viruses-15-01326-f002:**
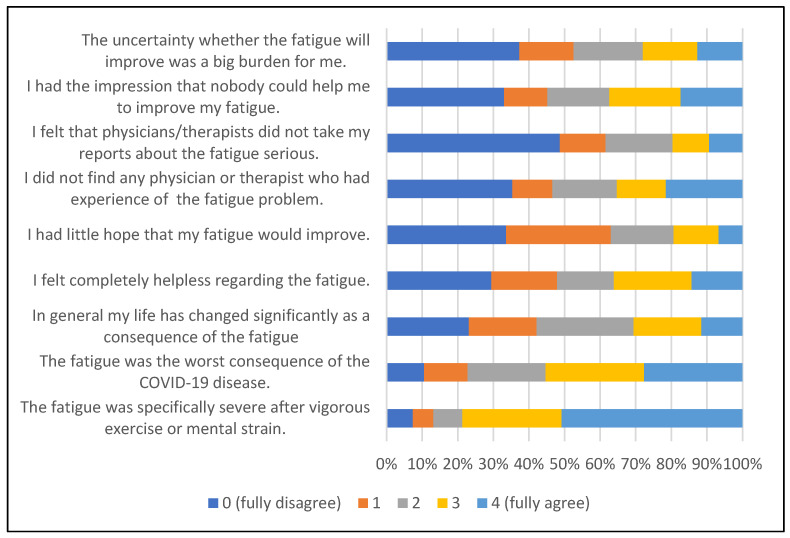
Experiences with fatigue lasting for at least 3 months after diagnosis (*n* = 123).

**Table 1 viruses-15-01326-t001:** Sample characteristics.

	Total (*n* = 304)		PCS * Yes (*n* = 210)		PCS * No (*n* = 94)		
	*n*	%	*n*	%	*n*	%	*p*-Value
Sex							0.0904
Male	127	41.8	81	38.6	46	48.9	
Female	177	58.2	129	61.4	48	51.0	
Age (median IQR)	53.5	41; 61	52.0	40; 59	52.0	39; 60	0.3455
Education							0.6278
≤9 years	52	17.1	40	19.1	18	19.1	
>9 years	252	82.9	170	80.9	76	80.9	
Living alone, yes	69	23.0	45	21.7	24	25.8	0.4388
Smoking							0.2792
Never a smoker	160	52.6	108	51.4	52	55.3	
Ex-smoker	123	40.5	90	42.9	33	35.1	
Current smoker	21	6.9	12	5.7	9	9.6	
Body Mass Index							0.0306
≤30 kg/m^2^	254	83.6	169	80.5	85	90.4	
>30 kg/m^2^	50	16.44	41	19.55	9	9.6	
Comorbidities							
Hypertension	64	21.1	41	19.6	23	24.5	0.3386
Diabetes	14	4.6	11	5.3	3	3.2	0.5611
Myocardial infarction	7	2.3	6	2.9	1	1.2	0.3328
Coronary artery disease	16	5.3	14	6.7	2	2.1	0.1622
Stroke	6	1.9	4	1.9	2	2.1	1
Anxiety disorder	18	5.9	15	7.2	3	3.2	0.0560
Chronic bronchitis	19	6.3	17	8.2	2	2.1	0.0771
Depression	27	8.9	23	11.0	4	4.3	0.028
Autoimmune disorder	28	9.2	20	9.6	8	8.5	0.8877
Cancer	15	4.9	10	4.8	5	5.3	0.6259
Recurrent COVID-19	73	24.6	54	26.5	19	20.4	0.5332
Time between first positive PCR test and follow-up survey							0.6679
>12 to ≤15 months	8	2.6	5	2.4	3	3.2	
>15 bis ≤18 months	13	4.3	10	4.8	3	3.2	
>18 bis ≤21 months	74	24.3	56	26.7	18	19.2	
>21 bis ≤24 months	52	17.1	32	15.2	20	21.3	
>24 bis ≤27 months	62	20.4	43	20.5	19	20.2	
>27 bis ≤30 months	94	30.9	63	30.0	31	33.0	
>30 months	1	0.3	1	0.5	0	0	
Median IQR	26	20.5; 27.2	25.9	20.2; 27.0;	26.0	20.7; 27.2	0.5134
Min/Max		14.1/30.2		14.1/30.2		14.2/29.8	
Post-COVID Functional Status							<0.0001
No limitations	219	73.0	129	62.0	90	97.8	
Negligible limitations	42	14.0	40	19.2	2	2.2	
Slight limitations	29	9.7	29	13.9	0	0	
Moderate limitations	10	3.3	10	4.8	0	0	
Severe limitations	0	0	0	0	0	0	

* Post-COVID-19 Syndrome.

**Table 2 viruses-15-01326-t002:** Symptoms at follow-up reported by study participants with or without post-COVID-19 Syndrome (PCS).

	PCS Yes (*n* = 210)		PCS No (*n* = 94)		
	*n*	%	*n*	%	*p*-Value
Fatigue or exhaustion	158	75.2	0	0	<0.0001
Muscle—or joint pain	110	52.4	17	18.1	<0.0001
Concentration problems	105	50.0	0	0	<0.0001
Headache	101	48.1	13	13.8	<0.0001
Memory problems	101	48.1	0	0	<0.0001
Sleepiness	93	44.3	0	0	<0.0001
Sleep problems	84	40.0	7	7.5	<0.0001
Dyspnea on exertion	82	39.1	0	0	<0.0001
Flatulence	82	39.1	12	12.8	<0.0001
Mood swings	78	37.1	2	2.1	<0.0001
Rhinitis or running nose	67	31.9	16	17.0	0.0079
Stuffy nose	66	31.4	14	14.9	0.0025
Depressive mood	59	28.1	2	2.1	<0.0001
Throat pain	57	27.1	10	10.6	0.0013
Palpitations	53	25.2	2	2.1	<0.0001
Cough	52	24.8	17	18.1	0.199
Muscle stiffness	46	21.9	4	4.3	<0.0001
Teary eyes	45	21.4	5	5.3	0.0003
Muscle weakness	45	21.4	0	0	<0.0001
Feelings of pins and needles in arms or legs	45	21.4	0	0	<0.0001
Vertigo	44	21.0	3	3.2	<0.0001
Impairment of smell function	42	20.0	8	8.5	0.0125
Diarrhea	38	18.1	8	8.5	0.0311
Swallowing pain	37	17.6	4	4.3	0.0010
Heartburn	36	17.1	9	9.6	0.0859
Chest pressure or pain	36	17.1	0	0	<0.0001
Anxiety, panic	35	16.7	1	1.1	<0.0001
Hair loss	30	14.3	2	2.1	0.0009
Impairment of taste function	29	13.8	6	6.4	0.0791
Stomach pain	28	13.3	3	3.2	0.0068
Impaired vision	28	13.3	2	2.1	0.0015
Problems with coordination of movements	27	12.9	0	0	<0.0001
Dyspnea on rest	25	11.9	0	0	<0.0001
Skin rash	19	9.1	3	3.2	0.0923
Loss of appetite	19	9.1	0	0	0.0013
Increased temperature	15	7.1	3	3.2	0.2916
Pink eyes or conjunctivitis	13	6.2	2	2.1	0.1604
Nausea or vomiting	12	5.7	1	1.1	0.0716
Shivering	12	5.7	0	0	0.0208
Fever (38.1 °C or higher)	12	5.7	0	0	0.0208
Blue lips	4	1.9	0	0	0.3151
Haemoptysis	0	0	0	0	-

**Table 3 viruses-15-01326-t003:** Health care utilization 4 weeks and 12 months before the follow-up survey in participants with and without post-COVID-19 Syndrome (PCS).

	Total (*n* = 304)		PCS Yes (*n* = 210)		PCS No (*n* = 94)		
	*n*	%	*n*	%	*n*	%	*p*-Value
Past 4 weeks							
Hospitalization	1	0.3	1	0.5	0	0	1.000
COVID outpatient clinic	1	0.3	1	0.5	0	0	1.000
Other outpatient clinic	1	0.3	1	0.5	0	0	1.000
Counseling center	2	0.7	2	1.0	0	0	1.000
General practitioner/internal medicine	62	21.2	48	24.0	14	15.2	0.0882
Specialist in gastro-intestinal diseases	6	2.2	6	3.2	0	0	0.1815
Specialist in gynecology	19	6.7	17	8.9	2	2.2	0.0419
Specialist in dermatology	9	3.2	9	4.7	0	0	0.0613
Specialist in ear, nose and throat	13	4.6	12	6.2	1	1.1	0.0690
Specialist in cardiology	14	4.9	12	6.2	2	2.3	0.2382
Specialist in neurology	8	2.9	8	4.2	0	0	0.0584
Specialist in ophthalmology	14	5.0	13	6.8	1	1.1	0.0720
Specialist in orthopedics	13	4.6	12	6.2	1	1.1	0.0693
Specialist in pulmonology	13	4.7	11	5.8	2	2.3	0.2369
Specialist in psychiatry/psychotherapy	12	4.3	12	6.3	0	0	0.0112
Specialist in urology	2	0.7	2	1.1	0	0	1.000
Occupational therapist	3	1.0	3	1.5	0	0	0.5545
Non-medical practitioner	6	2.0	5	2.5	1	1.1	0.6692
Osteopath	3	1.0	1	0.5	2	2.2	0.2317
Physical therapist	12	4.0	10	4.9	2	2.2	0.3539
Psychologist/psychotherapist	6	2.0	6	3.0	0	0	0.1818
Past 12 months							
Hospitalization	1	0.3	1	0.5	0	0	1.000
COVID outpatient clinic	10	3.3	10	4.8	0	0	1.000
Other outpatient clinic	3	1.0	3	1.4	0	0	1.000
Counseling center	5	1.7	4	2.0	1	1.1	1.000
General practitioner/internal medicine	125	42.7	101	50.5	24	25.8	<0.0001
Specialist in gastro-intestinal diseases	12	4.3	10	5.3	2	2.3	0.3494
Specialist in gynecology	31	11.0	25	13.0	6	6.7	0.1148
Specialist in dermatology	15	5.4	15	7.8	0	0	0.0037
Specialist in ear, nose and throat	20	7.1	19	9.8	1	1.1	0.0057
Specialist in cardiology	44	15.6	40	20.7	4	4.5	0.0005
Specialist in neurology	23	8.2	22	11.4	1	1.1	0.0019
Specialist in ophthalmology	18	6.4	15	7.8	3	3.4	0.1964
Specialist in orthopedics	24	8.5	22	11.3	2	2.3	0.0103
Specialist in pulmonology	41	14.5	37	19.1	4	4.5	0.0009
Specialist in psychiatry/psychotherapy	16	5.8	16	8.5	0	0	0.0020
Specialist in urology	7	2.5	5	2.6	2	2.3	1.000
Occupational therapist	3	1.0	3	1.5	0	0	0.5545
Non-medical practitioner	15	5.1	14	7.0	1	1.1	0.0428
Osteopath	15	5.1	13	6.4	2	2.2	0.1593
Physical therapist	23	7.7	21	10.2	2	2.2	0.0173
Psychologist/psychotherapist	11	3.7	11	5.5	0	0	0.0197

## Data Availability

The datasets generated during and/or analyzed during the current study are not publicly available due to data protection requirements but are available in an anonymized form from the corresponding author on reasonable request.

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
