# Peer review of "Post-COVID-19 Syndrome in Non-Hospitalized Individuals: Healthcare Situation 2 Years after SARS-CoV-2 Infection"

_viruses, 2023, doi:10.3390/v15061326_

Round 1
Reviewer 1 Report
The authors have made an interesting attempt at “Post COVID-19 Syndrome in Non-hospitalized Individuals: Healthcare Situation 2 Years after SARS-CoV-2 Infection.” The manuscript is interesting; however, the authors need to justify the scientific writing manuscript. Some of the general comments are provided below:
1. The study mentions that a total of 525 participants were enrolled, but only 361 participants returned a completed questionnaire in the follow-up survey. What factors might have influenced the lower response rate? Could non-response bias affect the generalizability of the findings?
2. How was healthcare utilization measured in the study? What specific aspects of health care utilization were assessed (e.g., doctor visits, hospitalizations, medication usage)? Were participants asked to recall and report their healthcare utilization over specific periods accurately?
3. The study mentions that the sample consisted of 177 women and 127 men. Was there any specific reason for the gender imbalance in the sample? Did the study analyze and report any gender differences in symptom burden, health care utilization, or experiences with PCS health care?
4. The study reports a median age of 53 years with an interquartile range (IQR) of 41 to 61 years. Were there any specific age groups that were overrepresented or underrepresented in the sample? How does the age distribution compare to the broader population affected by COVID-19 and PCS?
5. The study reports that at baseline, 245 participants (80.6%) reported at least one symptom, and at follow-up (FUP), 262 participants (86.2%) reported symptoms. Were these symptoms specifically related to COVID-19 or were they inclusive of any general symptoms experienced by the participants during the study period? Did the study differentiate between persistent COVID-19 symptoms and other unrelated symptoms?
6. The study states that participants with PCS had significantly higher prevalences in 33 out of 42 assessed symptoms at FUP. How were the PCS and non-PCS groups defined? Did the study control for confounding factors or adjust for demographic and clinical characteristics when comparing symptom prevalences between the two groups?
7. The study reports that a significant number of participants were unsatisfied with the information on PCS provided by various sources, such as the media, physicians/therapists, and health insurance companies/other healthcare providers. What specific aspects of the information were participants dissatisfied with? Did the study explore the reasons for dissatisfaction or participants' expectations regarding the information they received?
8. The study highlights that a significant proportion of participants who experienced fatigue for more than 3 months considered it specifically severe after vigorous exercise or mental strain. Were participants provided with any guidance or recommendations regarding managing fatigue with physical or mental exertion? Did the study investigate the impact of fatigue on participants' daily activities and quality of life?
9. The study provides information about the experiences and perceptions of participants within the specific sample studied. To what extent can the findings be generalized to a broader population? Did the study consider any potential biases or limitations that could affect the generalizability of the results?
Reviewer 2 Report
Please see the attachment.
